# Spatio-Temporal Differentiation and Driving Mechanism of the “Resource Curse” of the Cultivated Land in Main Agricultural Production Regions: A Case Study of Jianghan Plain, Central China

**DOI:** 10.3390/ijerph18030858

**Published:** 2021-01-20

**Authors:** Yuanyuan Zhu, Xiaoqi Zhou, Yilin Gan, Jing Chen, Ruilin Yu

**Affiliations:** Key Laboratory for Geographical Process Analysis & Simulation Hubei Province, Central China Normal University, Wuhan 430079, China; zhuyy990@mail.ccnu.edu.cn (Y.Z.); zhouxiaoqi@mails.ccnu.edu.cn (X.Z.); ganyilin@mails.ccnu.edu.cn (Y.G.); jing.chen@mail.ccnu.edu.cn (J.C.)

**Keywords:** resource curse, coefficient of cultivated land resource curse, kernel density estimation, regression model, panel data, China

## Abstract

Cultivated land resources are an important component of natural resources and significant in stabilizing economic and social order and ensuring national food security. Although the research on resource curse has progressed considerably, only a few studies have explored the existence and influencing factors of the resource curse of non-traditional mineral resources. The current study introduced resource curse theory to the cultivated land resources research and directly investigated the county-level relationship between cultivated land resource abundance and economic development. Meanwhile, the spatiotemporal dynamic pattern and driving factors of the cultivated land curse were evaluated on the cultivated land curse coefficient in China’s Jianghan Plain from 2001 to 2017. The results indicated that the curse coefficient of cultivated land resources in Jianghan Plain generally shows a downward trend. That is, the curse phenomenon of the cultivated land resources in large regions did not improve significantly in 2001–2017. The influencing factors of the cultivated land resource curse in different cursed degree areas varied and the spatial interaction of the cursed degree areas differed as well. This study proposed a transmission mechanism of the cultivated land resource curse in Jianghan Plain. Policies from throughout the entire and within the main agricultural producing areas were proposed to adjust the cultivated land resource curse. The results and conclusions of this study will be beneficial in improving future land-use policies in major agricultural areas and reducing lag in economic development caused by the strict protection of cultivated land resources.

## 1. Introduction

In the traditional economic growth theory, natural resources are essential to material production activities and a guarantee for economic growth and social development. However, many empirical studies have shown that most countries with abundant natural resources having a low economic development level [1,2,3,4]. The existing literature shows evidence that abundant natural resources are negatively correlated with regional economic growth. Furthermore, an essential concept of resource economics, the “resource curse,” has emerged [5]. However, this negative correlation has not been conclusively determined [6,7,8]. The abundance of natural resources does not have to be a negative factor but can also allow these regions to continue to benefit from natural resources. For example, Norway and Botswana have successfully turned the curse of natural resources into a blessing [9]. Thus, obtaining reliable empirical results to understand the relationship between natural resources and economic growth would be interesting. China is the largest developing country, and its economy needs further development. China’s natural resources are also diversified and abundant. Therefore, exploring the linkage relationship between economic development and natural resource abundance is of great significance, and empirical investigations of the relationship between resource abundance and economic growth provide new insights for policymakers to use natural resources as a blessing rather than a curse.

Cultivated land resources are an essential component of the natural resources and a necessary material condition for stabilizing economic and social order and guaranteeing food and ecological security [10]. China’s total cultivated land resources are relatively large and rank high globally. However, China’s per person cultivated land area is below 40 percent of that of the world [11]. After 1978, China’s reform and opening-up resulted in rapid economic and social development. Note that the rapid urban scale expansion has led to a rapid increase in construction and industrial lands and a rapid decrease in cultivated land. To ensure the country’s food security, the Chinese central government has formulated and promulgated various policies to protect cultivated land [12]. After implementing a series of policies, the cultivated land resources in China’s main agricultural areas have been protected. However, development has been restricted, and the rapid transformation of cultivated land to other types of land has been curbed [13]. Consequently, “how to continue to develop rapidly in the main agricultural areas under the strict cultivated land protection policy” has become a new problem. This issue is one of the reasons that this study focused on the relationship between the abundance of cultivated land resources and economic growth. Previous research on “resource curse” has generally focused on traditional consumption resources [14], such as gold [15,16], coal [17,18], and oil [19,20]. Thus, whether a relationship exists between arable land resources (as a non-traditional consumption resource) and economic growth is also worth exploring. From the “resource curse” perspective, this research uses Jianghan Plain as an example to study the relationship between cultivated land resource abundance and economic growth. Moreover, the current study provides timely and significant policy suggestions for the economic development of major agricultural producing areas.

The rest of the paper is organized as follows. Section 2 reviews the existing literature on the “resource curse” in detail and puts forward its own opinions. Section 3 introduces the study area’s general situation, clarifies why this study area was chosen and gives the research method to analyze the existence, spatial distribution, and mechanism of the “curse of cultivated land.” The results obtained in this study are explained in detail in Section 4. Section 5 puts forward targeted suggestions based on the research results, points out the shortcomings of this study, and draws the research conclusions of this paper. 

## 2. Literature Review

The “resource curse” means that abundant natural resources will hinder economic growth under certain circumstances. Gelb [21] used the word “curse” for the first time when he assessed the impact of huge profits from oil resources on six developing oil-producing countries. Auty [5] proposed the concept of “resource curse” for the first time when he studied mineral producing countries’ economic development. He believed that the economic development of natural resource-rich economies lag behind the relatively resource-poor economies. Empirical studies on the resource curse effect have been carried out successively. Most such studies have primarily looked into the relationship between the traditional consumption of natural resources and economic growth [22,23,24]. Sachs and Warner [25] are the first scholars to conduct empirical research on the “resource curse.” Their results have shown that the negative relationship between resource endowment and economic growth remains genuine even if they control variables that are important to economic growth. Plenty of studies in the existing literature have explained the transmission mechanism of the “resource curse” effect, mainly including the “Dutch disease” effect, “crowding out” effect, and “weakening system” effect [26]. For example, based on the “Dutch disease” effect, Usui and Avalos [27,28] explored the transmission mechanism of the “resource curse” effect in Papua New Guinea, Mexico, and Indonesia. They believed that these countries should strengthen macroeconomic management to avoid attracting excessive labor and capital from the agriculture and manufacturing sectors to the resource sector. Li and Morrisetal [29,30] concluded that resource-based areas’ economic structure is unreasonable; the “locking” and “crowding out” effects hindered the upgrading of the local industrial network. Goel, Zhan, and Tyburski [31,32,33] tested whether the corruption caused by abundant resources is the root cause of the resource curse. However, not all countries have the same situation. Weber [34] found little evidence that increased gas production in the first decade of the 21st century threatened some counties in the south-central United States. Obeng-Odoom [35] found that the Oil City was not a single blessed or cursed city but rather a competitive arena in which curses and blessings coexist. Furthermore, with the deepening of the research, some researchers have begun to use the “resource curse” perspective to analyze the relationship between tourism resources, marine resources, and other non-traditional consumption natural resources and economic growth [14,36].

In China, many empirical studies have been carried out at the national and provincial levels [37]. For example, Xu and Wang [38] selected raw coal, crude oil, and pig iron as representatives of natural resources to study. Their study’s results verified that China also has a “resource curse” effect for the first time. Taking oil, gas, and mineral resources as research objects, Zhang, Gu, and Guo [39,40,41] quantitatively studied the internal relationship and temporal and spatial differentiation between resource abundance and economic growth. These studies analyzed the existence of the resource curse under different research scales and put forward the optimal path and policy suggestions for resource-based economic transformation. Simultaneously, a few studies have analyzed the transmission mechanism of the “resource curse” effect from many aspects, such as material capital input, human capital input, and technological innovation input [42,43,44,45]. Most studies in China have concluded a resource curse, especially in the country’s fast-growing central and western regions, but some have found a different story [46]. For example, Jing [47] found that China has no resource curse problem based on mineral resources in 31 Chinese provinces, and Deng and Wang [48] concluded that whether China has a resource curse problem is uncertain.

Through the literature review, we get the development history of the resource curse research (as shown in Figure 1). Despite the tremendous progress in researching the “resource curse,” some limitations still exist. First, the research objects are mostly concentrated on coal, oil, gold, and other traditional natural resources. Second, the research areas are mostly focused on the macro-level [37,39]. However, the larger spatial scale is easy to ignore the smaller regional category’s spatial heterogeneity, thus covering up the role of resource abundance on economic growth. Given this, this study attempted to expand the research field of the resource curse. We studied the relationship between cultivated land resources, which is a non-traditional natural resource of consumption, and economic growth. Simultaneously, the present study carried out the analysis from the relatively microscopic county-level scale and considered the spatial spillover effect to establish the model for analyzing the transmission mechanism.

## 3. Materials and Methods

### 3.1. The Connotation between Abundant Cultivated Land Resources and Economic Development

According to the current research results, we obtained the connotation between abundant cultivated land resources and economic development (as shown in Figure 2). Abundant cultivated land resources interacted with economic development through various driving factors. The positive correlation was “cultivated land resource blessing,” and the negative correlation was “cultivated land resource curse.” The policy enlightenment was given by analyzing the transmission mechanism between abundant cultivated land resources and economic development. The cultivated land resource blessing still existed, and the cultivated land resource curse completed the transformation from “curse” to “blessing.”

### 3.2. Study Area and Data Sources

Jianghan Plain, which is located in the south-central Hubei Province, is an integral part of the Plain of the Yangtze River’s middle and lower reaches. It is situated in latitude 29°26′–31°36′ N and longitude 110°14′–114°13′ E, covering an area of approximately 38,000 square kilometers (as shown in Figure 3). Jianghan Plain covers eleven cities, six counties, and one district (all county-level administrative regions). Jianghan Plain has excellent hydrothermal conditions, flat terrain, fertile soil, and rich cultivated land resources. As an essential grain, cotton, and oil production base in China, Jianghan Plain has always undertaken the vital mission of ensuring agriculture and food security. Given its unique geographical advantages, Jianghan Plain was once one of China’s most economically developed areas [49] and was reputed as “Home of Fish and Rice.” However, in recent decades, the economic development of the Jianghan Plain has obviously lagged, with the proportion of its GDP in Hubei Province gradually decreasing from 25.76% in 1990 to 16.72% in 2018. The proportion of its GDP in the whole country gradually decreased from 1.18% in 1990 to 0.71% in 2018 (Hubei Provincial Bureau of Statistics).

The data used in this study were based on the analysis and collation of the Hubei Statistical Yearbook [50], the Hubei Rural Statistical Yearbook [51], and the China County Statistical Yearbook [52].

### 3.3. Research Methodology

#### 3.3.1. Coefficient of the Cultivated Land Resource Curse

The resource curse means that abundant natural resources may be a curse rather than a blessing for economic development. To quantify the degree of the curse of cultivated land resources in the region, we drew lessons from the current research results [14,37] and used the ratio of regional resource endowment to economic development to reflect the deviation between local economic development and resource advantage. Given the difficulty in quantifying factors, such as the fertility degree of cultivated land resources, we used the commonly used cultivated land area to reflect the abundance of cultivated land resources. We also used the regional GDP to express the degree of economic development. Thus, the formula for calculating the curse coefficient of cultivated land resources is as follows:(1)$$\;ES_{i}=\frac{E_{i}/\sum_{i=1}^{n}E_{i}}{SI_{i}/\sum_{i=1}^{n}SI_{i}}$$
where *ES_i_* represents the curse coefficient of cultivated land resources in area *i*, n represents the number of areas (18 in this study), *E_i_* represents the cultivated area of area *i*, and *SI_i_* represents region *i*’s gross product. If the coefficient is larger, the cultivated land resource curse is more serious. If the coefficient is less than 1, then no cultivated land resource curse phenomenon is observed in this area.

The classification standard of the cultivated land resources curse degree was adopted in this paper based on the research of “Chinese regional differences and its driving force analysis of the resource curse” [53]. According to the results, based on the deviation degree of economic advantage and resource advantage, the classification of resource curse degree can be divided into four types. The criteria of the cultivated land resource curse effect type classifications are shown in Table 1.

#### 3.3.2. Kernel Density Estimation (KDE)

To analyze the temporal evolution of the cultivated land resource curse in Jianghan Plain, we used Stata15 (StataCorp., College Station, TX, USA). to conduct the KDE of the curse coefficient of cultivated land resource in Jianghan Plain. The principle of KDE is to calculate the number of points around a point. If the data is *x*_1_, *x*_2_,…, *x_n_*, a kernel density at any point *x* is estimated as follows:(2)$$f_{h}\left(x\right)=\frac{1}{nh}\overset{n}{\underset{i=1}{\sum }}K\left(\frac{x-x_{i}}{h}\right)$$
where *K*(•) is the kernel function, which satisfies the symmetry and the integral of *K*(*x*)d*x* = 1; *x*_1_, *x*_2_,…, *x_n_* are identically distributed independent samples extracted from the population with distribution density function *f* to estimate the value of *f* at a certain point; (*x* − *x_i_*) represents the distance from the estimated point to sample *x_i_*; *h* is called bandwidth, which affects the shape and smoothness of the curve. 

#### 3.3.3. Model Set for the Factors Driving the Cultivated Land Resource Curse

##### Selection of Driving Factors

Based on the availability of relevant regional statistical data and the differences between cultivated land resources and traditional energy resources, this study drew on the existing research results [54,55] and constructed the driving factors for the curse of cultivated land resources at the county level in China from the following seven aspects:(1)Material capital investment level: Material capital is an important driving force of economic development. This study used the entire society’s fixed-asset investment per hectare of land (*FI*) to measure this index. The larger the index is, the greater the investment in infrastructure in the region, which is conducive to developing various industries;(2)Regional population’s status: The relationship between regional population status and economic development is close. This study used the urbanization rate (*UR*) to measure this index. The larger this index is, the more people are concentrated in cities to engage in high-value-added industries;(3)Cultivated land protection policy: The cultivated land protection policy is a basic national policy in China. It makes it difficult for areas with abundant arable land resources to be transformed into other land types. This study selected the per capita cultivated land area (*PCA*) to measure this index;(4)Industrial upgrading level: The local industrial structure reflects the composition, connection, and proportion of each industry’s regional economic system. This study selected the proportion of the output value of secondary and tertiary industries in GDP (*STGD*) as a model factor;(5)Agricultural intensification level: The intensification of agriculture makes the best possible use of the cultivated land per unit area. This study used the gross agricultural product per hectare of land (*GA*) and the grain output per hectare of land (*GO*) as indicators to measure the level of local agricultural intensification;(6)Agricultural mechanization level: Agricultural mechanization is an important agricultural infrastructure. This study selected the total power of machinery on land (*TPM*) to reflect the level of local agricultural mechanization;(7)Spatial correlation: Regional economic development is not independent of each other but influences each other. This study considered the spatial correlation and introduced the resource curse coefficient’s first-order spatial lag term as an independent variable. It can be expressed as follows:
(3)$$Y_{1}=Wy_{t,i-1}$$
where *Y*_1_ represents the resource curse space coefficient of the first-order lag, $y_{t,i-1}$ represents the cross-sectional data of the resource curse coefficient of each region with a lag of one period, and *W* represents the spatial weight matrix. If two counties are adjacent in geographical location, they are recorded as 1 in *W*; otherwise, 0. We used this formula in Stata15 to calculate for subsequent regression analysis.


##### Regression Model of Driving Factors

Based on the existing research results [56,57,58], a regression analysis model was selected to study the driving factors of the curse of cultivated land resources. To simplify the processing, this study drew on the ideas of Liu et al. [59], introduced the autocorrelation coefficient into general panel regression based on the spatial lag model, and adopted the following model:(4)$$y_{t,i}=\rho Wy_{t,i-1}+\alpha x+\varepsilon_{t,i}$$
where *t* represents the year, *i* corresponds to the cross-sectional data of each region, $\;y_{t,i}$ represents the cross-sectional data of the resource curse coefficient of each region, *ρ* represents the spatial autocorrelation coefficient, *W* represents the spatial weight matrix, $y_{t,i-1}$ represents the cross-sectional data of the resource curse coefficient of each region with a lag phase, *α* represents the coefficient of the variable, and *ε* represents the error term.

By substituting the variables selected above, this study established the regression model as follows:(5)$$y_{t,i}=\alpha_{1}FI+\alpha_{2}UR+\alpha_{3}PCA+\alpha_{4}STGD+\alpha_{5}GA+\alpha_{6}GO+\alpha_{7}TPM+\alpha_{8}Y_{1}+\varepsilon_{t,i}$$
where *t* represents the year, *i* corresponds to the cross-sectional data of each region, $y_{t,i}$ represents regional cultivated land resource curse the coefficient of cross-sectional data, *FI* represents the level of material capital investment, *UR* represents the regional population’s status, *PCA* represents the protection policy of cultivated land, *STGD* represents the industrial upgrading level, *GA* and *GO* represents the agricultural intensification level, *TPM* represents the level of agricultural mechanization, $Y_{1}$ represents spatial correlation, α represents the coefficient of the variable, and ε represents the error term. 

In this study, we imported independent and dependent variable data into Stata15, performed balance panel and standardization processing. Next, we used Eviews10 to run the spatial recursive panel data model for the cultivated land resource curse and related variables. Finally, we performed the unit root test and co-integration test on each variable. In software operation, there are three types of panel data models. We decided which model to choose according to the residual sum of squares of the variable coefficient model, the variable intercept model, and the mixed model.

## 4. Results

### 4.1. Temporal and Spatial Evolution of the Cultivated Land Resource Curse in Jianghan Plain

#### 4.1.1. Temporal Evolution of the Cultivated Land Resource Curse in Jianghan Plain

The curse coefficients of cultivated land resources were calculated using Equation (1). The results are shown in Table 2. From 2001 to 2017, the curse coefficient of cultivated land resources in the Jianghan Plain in China showed a downward trend. The coefficient declined significantly between 2001 and 2005 (from 2.194 in 2001 to 1.102 in 2005, with an average of 12.45%) and slightly increased from 2005 to 2012 (from 1.102 in 2005 to 1.195 in 2012). Since then, it had been relatively stable, fluctuating slightly between 1.19 and 1.20. However, the cultivated land resource curse coefficients for Zhijiang, Zhongxiang, Shishou, Yingcheng, Xiantao, and Gongan showed significant fluctuations from 1.093, 1.018, 0.743, 0.525, 0.892 and 1.682 in 2001 to 0.382 and 1.534, 1.452, 0.834, 0.610, and 1.914 in 2017, respectively. The curse coefficients for Dangyang, Shayang, Jiangling, Jianli, and Honghu had no fluctuation. In terms of the dispersion between 2001 and 2017, the variance of Jiangling and Shishou exceeded 0.05. The curse coefficients of cultivated land resources underwent considerable fluctuation over 17 years, and the degree of data dispersion was high. The curse coefficients of Caidian, Shayang, Zhongxiang, Yingcheng, Jianli, Honghu, and Songzi underwent a little fluctuation over 17 years. The data were more discrete, and the variance ranged from 0.01 to 0.05. The variance of curse coefficients of cultivated land resources in other areas was less than 0.01, and the fluctuation was smaller.

The results of KDE analysis of resource curse coefficients in 2001, 2009, and 2017 are shown in Figure 4. According to the analysis of the curve’s position, from 2001 to 2017, the curve moved first to the left and then to the right, but the change was unnoticeable. Based on the analysis of the shape of the curves, we found that the core density maps of the cultivated land resource curse in three years were a single peak distribution, and the overall span of the curve had no noticeable change. It showed that the curse of cultivated land resources in Jianghan plain had not been improved obviously. The peak value of the curve was the most significant in 2001 and gradually showed a downward trend with time. Given that the peak value areas were in a low curse coefficient position, the curse phenomenon of cultivated land resources in a large area had become more pronounced from 2001 to 2017.

#### 4.1.2. Spatial Evolution of the Cultivated Land Resource Curse in Jianghan Plain

According to the threshold criteria for the various resource curse zones (Table 1) and the cultivated land resource curse coefficients in Jianghan Plain, the year 2001, 2009, and 2017 were selected as the temporal nodes. The geographic information system (GIS) spatial analysis technology was used to visually analyze the pattern of the spatial evolution of the cultivated land resource curse in Jianghan Plain. The results are shown in Figure 5. Overall, the spatial distribution of cultivated land resource curse had changed to some extent from 2001 to 2017, with significant differences from 2001 to 2009 and from 2009 to 2017. After constant changes, the spatial distribution at all levels in 2017 was similar to that in 2001.

The severe cultivated land resource curse zone had remained unchanged. This zone consisted of Jiangling and Jianli. The size of the slightly cultivated land resource curse zone first decreased and then increased from eight regions (Zhongxiang, Jingshan, Tianmen, Shayang, Zhijiang, Songzi, Gongan, and Honghu) in 2001 to six regions (Tianmen, Shayang, Honghu, Songzi, Gongan, and Shishou) in 2009 and eight regions (Zhongxiang, Jingshan, Tianmen, Shayang, Shishou, Songzi, Gongan, and Honghu) in 2017. The size of the potential cultivated land resource curse zone increased first and then decreased from four regions (Shishou, Qianjiang, Xiantao, and Hanchuan) in 2001 to five regions (Zhongxiang, Jingshan, Yingcheng, Hanchuan, and Xiantao) in 2009 and one region (Yingcheng) in 2017. The size of no cultivated land resource curse zone gradually increased. The area included Dangyang, Yunmeng, Yingcheng, and Caidian in 2001; Yunmeng, Caidian, Dangyang, Zhijiang, and Qianjiang in 2009; and Dangyang, Zhijiang, Qianjiang, Xiantao, Hanchuan, Caidian, and Yunmeng in 2017.

The specific changes of the cultivated land resource curse in the Jianghan plain of China can be summarized into four types: (1) Counties with insignificant changes in the curse of cultivated land resources included two counties with the severe curse (Jiangling and Jianli), five counties with a mild curse (Songzi, Gongan, Tianmen, Shayang, and Honghu), and three counties without curse (Yunmeng, Caidian, and Dangyang); (2) The cultivated land resource curse fluctuated and moved from the lightly cursed zone to the potentially cursed zone and then back to the lightly curse zone in two counties (Zhongxiang and Jingshan); (3) It improved in two counties (Yingcheng and Shishou); (4) The cultivated land resource curse deteriorated in four counties (Hanchuan, Xiantao, Qianjiang, and Zhijiang).

### 4.2. Driving Factors of the Cultivated Land Resource Curse in Jianghan Plain

Calculated the residual sum of squares of the variable coefficient model, the variable intercept model, and the mixed model, we get *F*1 = 3.8037 and *F*2 = 20.7311, which were larger than the critical values 1.3211 and 1.3121, respectively, in the 0.05 confidence interval. Thus, we rejected the variable intercept model and the mixed model and selected the variable coefficient model (the fitting degree was 98.69%). The results are shown in Table 3.

Based on the existing references [47], when we analyzed the variable coefficient model, we believed that as long as an index in one county passed the correlation test, then the index in the county’s sub-district also passed the test. If an index that passed the correlation test in a cursed zone showed both positive and negative correlation, the index was discarded in this zone. Therefore, the regression analysis results of driving factors for each curse zone are shown as follows:No cultivated land resource curse zone: The curse coefficient presented a significant negative correlation with the variable *Y*_1_, which represented the spatial correlation;Potential cultivated land resource curse zone: The curse coefficient of the cultivated land resource in each county was negatively correlated with *Y*_1_. A significant positive correlation was observed between *PCA* and curse coefficient. A significant negative correlation between *STGD*, *GA*, *GO*, and *TPM* and the curse coefficient of cultivated land resources were observed;Slightly cultivated land resource curse zone: The curse coefficient of cultivated land resources in each county was positively correlated with *Y*_1_ and passed the general significance test. The curse coefficient of each county was significantly positively correlated with *FI*; generally significantly positively correlated with *STGD*, *PCA*, and *UR*; and generally significantly negatively correlated with *TPM*;Severe cultivated land resource curse zone: The resource curse coefficient of this zone was positively correlated with *FI*, *Y*_1_, *UR*, *PCA*, and *STGD* but negatively correlated with *GA* and *TPM*.

## 5. Discussion

### 5.1. Transmission Mechanism of the Cultivated Land Resource Curse in Jianghan Plain

The regional economy’s rapid development will drive the upgrade of the population and society, thereby promoting regional economy and land use [60,61]. Our study demonstrated variety in the driving factors affecting the curse degree of the cultivated land resources in Jianghan Plain counties.

In the non-cultivated land resource curse zone, the curse coefficient presented a significant negative correlation with other adjacent areas. The cultivated land resources in these counties have been extensively used and transformed into high economic value. These areas play a prominent central position in economic development and have a large centripetal force effect [62,63], which substantially pulls on the surrounding economy and makes many labor, capital, and technology neighboring areas [64]. Consequently, the cultivated land resources in the neighboring areas have not been effectively developed, and the curse of cultivated land resources has intensified.

A negative correlation exists, although not evident, between the curse coefficient and other adjacent areas in the potential cultivated land resource curse zone. The curse coefficient was negatively correlated with *STGD*, *GA*, *GO*, and *TPM*. On the one hand, there is a centripetal force effect, although not evident, on the labor, capital, and technology of the surrounding areas. On the other hand, industrial upgrading, agricultural intensification, and mechanization have a strong pulling effect on regional economic development [65,66].

In the slightly cultivated land resource curse zone, the curse coefficient was positively correlated with other adjacent areas’ curse coefficient. In general, the curse coefficient was significantly positively correlated with *STGD*, *PCA*, and *UR*. The centrifugal effect of cultivated land resource curse on capital, technology, and labor force resulted in an evident outflow trend. The rapid development of the regional economy was not easily realized. 

In the severe cultivated land resource curse zone, the curse coefficient is positively correlated with *FI*, *Y*_1_, *UR*, *PCA*, and *STGD*; negatively correlated with *GA* and *TPM*. On the one hand, the substitution effect of technological progress, reduction of resource flow cost, and resource utilization efficiency aggravated resource advantage loss. Such an effect made the centrifugal effect on Jianli and Jiangling increasingly strong in the neighboring developed regions. On the other hand, the two counties’ agricultural intensification and mechanization were insufficient, the material capital investment was insufficient, and the industrial structure was relatively single.

We can obtain the transmission mechanism of the cultivated land resource curse in Jianghan Plain by further analyzing the above results, as shown in Figure 6. The spatial effect of different curse zones was different. The zones with potential cultivated land resource curse and no cultivated land resource curse had relatively large agglomeration benefits. These regions exerted centripetal force on capital, technology, labor force, and other factors of surrounding areas. In contrast, the zones with slightly cultivated land resource curse and severe cultivated land resource curse were not strong enough to transform into an economic advantage. These regions formed centrifugal force on capital, technology, labor force, and other factors, driving these factors to the zone with potential cultivated land resource curse and no cultivated land resource curse. Simultaneously, the interaction of centripetal force and centrifugal force made the gathering force of the non-cursed zone stronger and the economy more developed. The spillover effect of the cursed zone was more pronounced, and the weak position of the regional economy in the broader region was more prominent.

However, the non-cursed zone also had spillover effects. The high concentration of all elements made it overburdened. It caused environmental pollution and the overloading of public service facilities, which required some industries to be transferred from the no-curse zone. Therefore, the implemented and differentiated policies on the use of cultivated land resources according to regions with different curse degrees re-acted to the regional economy’s development and thus eliminated the curse of cultivated land resources within the region.

### 5.2. Policy Implications of the Cultivated Land Resource Curse

Our findings have a relative enlightening significance in implementing cultivated land use policy and economic development in China’s main agricultural producing areas. From the point of view of the entire main agricultural production areas, the protection of cultivated land aims to realize sustainable economic and social development, rather than simply restrict the occupation of cultivated land and restricting regional economic development. The implementation of cultivated land protection policy requires saving the land, improving land-use efficiency, and increasing land input and yield. First, financial support for agriculture must be increased. We should not ask major agricultural production areas to bear the cost of food production without any cost. The division of labor between the central and local governments in food security should be clearly defined to bear the national food security’s main cost. Second, through comprehensive agricultural development and irrigation and water conservancy construction, we will improve agricultural production conditions, improve the quality of arable land, increase land fertility, and build permanent high-standard farmland with stable and high yields. Third, agricultural science and technology investments should be increased and scientific and technological progress to agricultural production must be improved. In particular, farmers should be guided in developing big agriculture, implementing specialized division of labor, applying advanced technology and good varieties to improve production efficiency and effect, and developing characteristic agriculture and seeking benefits with products, lastly, farmers must be encouraged to transfer from rural areas to cities to revitalize the space for rural and agricultural development.

From the present agricultural main producing areas of the curse of arable land resources in each zone, each region should have different development measures. In the zone with no cultivated land resource curse, each region should maintain cultivated land resources’ development intensity. Each region should promote the rationalization and upgrading of industrial structure; vigorously develop high-tech industries; disperse resources and labor-intensive industries to surrounding areas; and stimulate the economic development of urban areas with labor, capital, and technology. In the zone with the potential cultivated land resource curse, each region should improve cultivated land resources’ utilization efficiency and appropriately relax cultivated land protection policies. This zone should take a new road of industrialization development, promote industrial upgrading, and drive economic development. Simultaneously, the zone with no cultivated land resource curse belongs to the destination of capital, technology, labor, and other transfer factors, promoting economic development by optimizing productivity distribution. In the zone with a slightly cultivated land resource curse, each region should optimize the investment environment; strengthen the construction of urban infrastructure; improve people’s mobility, logistics, and information flow; attract talents, capital, and other factors; and drive the development of the local economy. In the zone with a severe cultivated land resource curse, the transformation and upgrading of agriculture should be imperative. This area can adopt agricultural industrialization funds, support agricultural product processing technology to introduce and develop energetically, accelerate agricultural industrialization, and drive economic development.

## 6. Conclusions

The efficient utilization of cultivated land resources is a critical problem in cultivated land resource utilization and local economic development. This study used the resource curse as a basis to propose the “curse of cultivated land resources.” Thereafter, this research examined the dynamic changes of cultivated land resource curse coefficient and performed panel regression to determine the factors influencing cultivated land resource curse in 2001–2017 in Jianghan Plain, which is an agricultural production area in China. From 2001 to 2017, the curse coefficient of cultivated land resources in Jianghan Plain generally showed a downward trend, whereas the curse phenomenon of the cultivated land resources in large regions did not improve significantly. The influencing factors of the cultivated land resource curse in the different cursed degree areas were different, and the spatial interaction of the cursed degree areas was also different. This research proposed a transmission mechanism of the cultivated land resource curse in Jianghan Plain. Zones with potential cultivated land resource curse and non-cultivated land resource curse had relatively large agglomeration benefits. These regions exerted centripetal force on the capital, technology, labor force, and other factors of the surrounding areas. Other areas were affected by centrifugal force. The interaction of the centripetal and centrifugal forces substantially strengthened the gathering force of the non-cursed zone and considerably developed the economy. Moreover, the non-cursed zone exhibited spillover effects. Policies from throughout the entire and within the main agricultural producing areas, such as those related to increasing financial support for agriculture and strengthening comprehensive agricultural development, were proposed to adjust the cultivated land resource curse.

This work has the following limitations. The research time scale of this study was from 2001 to 2017 because of the availability of data. Whether the influencing factors selected in the transmission mechanism of cultivated land resources and economic development are perfect remains to be considered. Given the long-term nature of the curse effect of cultivated land, further research should select a longer time scale for analysis. The selection of variables between cultivated land resources and economic development should be further considered to enhance the persuasive power of revealing the curse effect of cultivated land resources.

## Figures and Tables

**Figure 1 ijerph-18-00858-f001:**
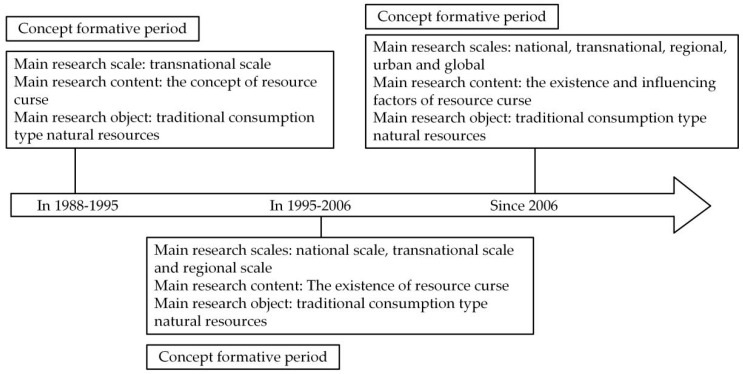
The development history of the resource curse research.

**Figure 2 ijerph-18-00858-f002:**
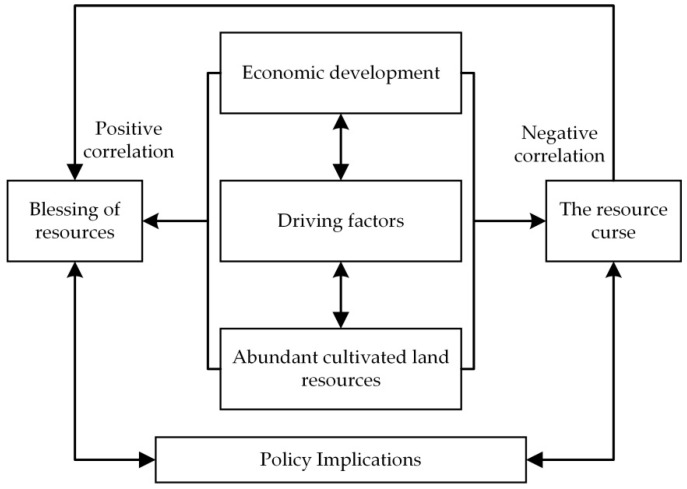
Connotation between abundant cultivated land resources and economic development.

**Figure 3 ijerph-18-00858-f003:**
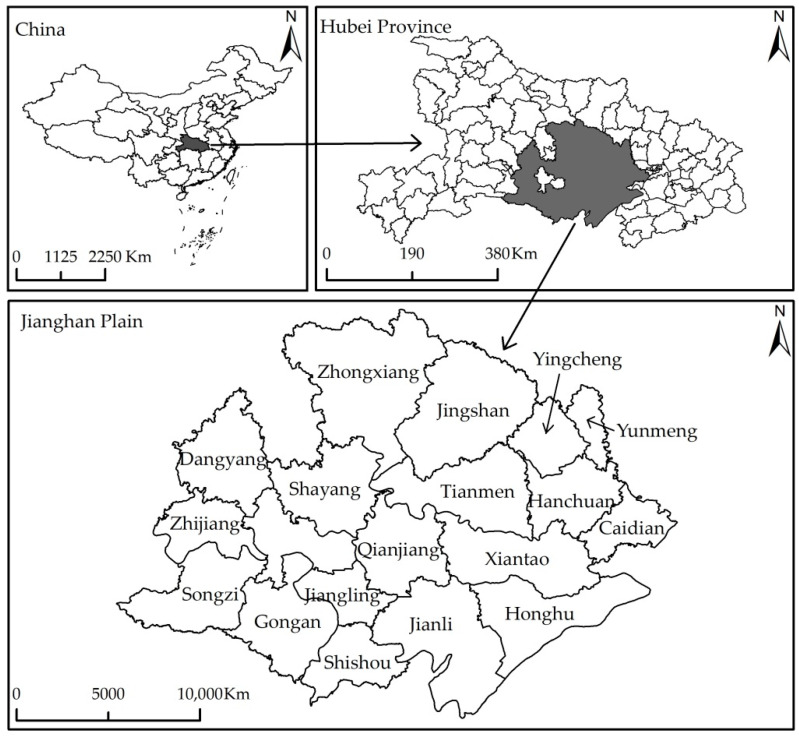
The location of the study area.

**Figure 4 ijerph-18-00858-f004:**
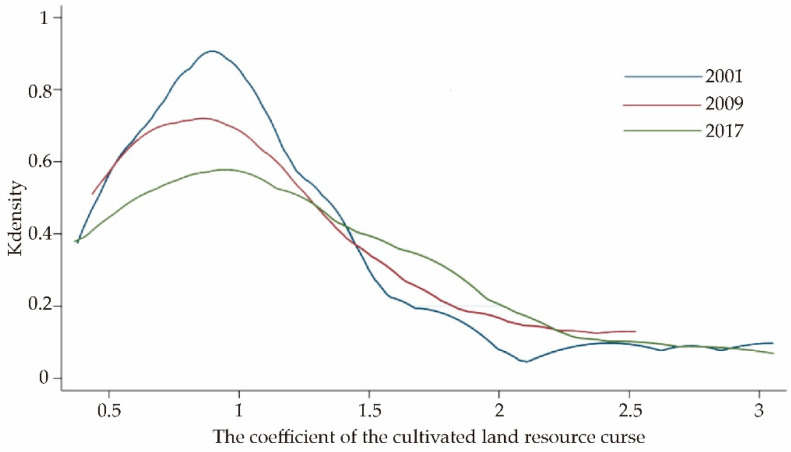
Kernel density distribution of the cultivated land resource curse coefficients for the county-level administrative regions in Jianghan Plain.

**Figure 5 ijerph-18-00858-f005:**
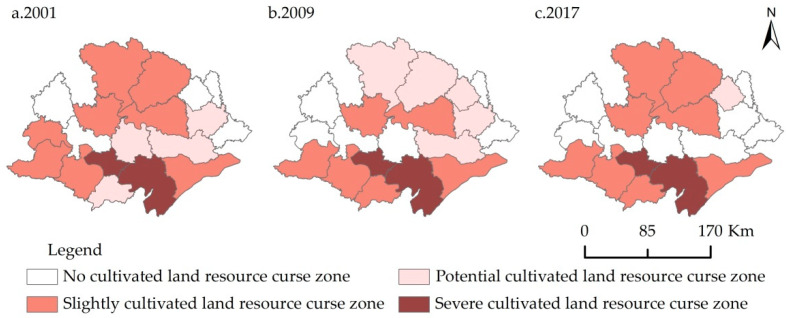
Spatial evolution of the cultivated land resource curse in Jianghan Plain.

**Figure 6 ijerph-18-00858-f006:**
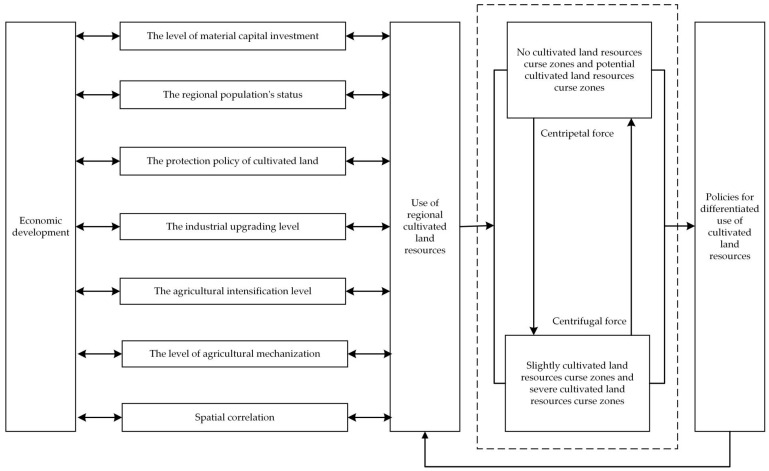
The transmission mechanism of the cultivated land resource curse in Jianghan Plain.

**Table 1 ijerph-18-00858-t001:** Zoning thresholds and basic characteristics of the cultivated land resource curse.

Zone	Threshold	Characteristics
No cultivated land resource curse zone	$ES_{i}<0.7$	The economy develops faster than the development determined by cultivated land resource endowments. No risk of developing cultivated land resource curse is observed.
Potential cultivated land resource curse zone	$0.7\leq \;ES_{i}<1$	The economy develops faster than the development determined by cultivated land resource endowments. A risk of developing a slightly cultivated land resource curse is observed.
Slightly cultivated land resource curse zone	$1\leq ES_{i}<2$	The cultivated land resource curse starts manifesting. It is at a low level. A risk of developing a more severe cultivated land resource curse is observed.
Severe cultivated land resource curse zone	$ES_{i}\geq 2$	The cultivated land resource advantage does not fully transform into an economic advantage.

**Table 2 ijerph-18-00858-t002:** Coefficients of the cultivated land resource curse for the county-level administrative regions in Jianghan Plain.

	2001	2003	2005	2007	2009	2011	2013	2015	2017	Mean	Variance
Caidian	0.379	0.718	0.565	0.501	0.436	0.363	0.325	0.307	0.367	0.429	0.017
Dangyang	0.596	0.652	0.799	0.780	0.648	0.562	0.623	0.616	0.597	0.653	0.005
Zhijiang	1.093	0.883	0.771	0.760	0.688	0.595	0.540	0.510	0.382	0.694	0.035
Jingshan	1.137	1.074	1.030	1.003	0.869	1.007	1.045	1.087	1.190	1.048	0.005
Shayang	1.652	1.590	1.424	1.269	1.594	1.538	1.605	1.677	1.651	1.564	0.011
Zhongxiang	1.018	1.035	1.071	1.031	0.987	0.986	1.013	1.062	1.534	1.053	0.015
Yunmeng	0.662	0.633	0.589	0.585	0.562	0.591	0.595	0.623	0.583	0.613	0.002
Yingcheng	0.525	0.521	0.734	0.771	0.792	0.768	0.765	0.775	0.834	0.711	0.012
Hanchuan	0.735	0.684	0.760	0.792	0.772	0.803	0.767	0.759	0.692	0.752	0.001
Jiangling	3.052	3.158	2.201	2.335	2.524	2.770	2.803	2.671	3.055	2.714	0.082
Gongan	1.682	1.594	1.686	1.716	1.733	1.822	1.848	1.838	1.914	1.753	0.009
Jianli	2.423	2.378	2.296	2.575	2.486	2.682	2.866	2.883	2.397	2.598	0.039
Shishou	0.743	0.692	0.817	0.891	1.005	1.236	1.361	1.337	1.452	1.055	0.075
Honghu	1.153	1.146	1.312	1.389	1.397	1.594	1.593	1.569	1.182	1.394	0.030
Songzi	1.247	1.198	1.409	1.502	1.454	1.436	1.349	1.310	1.290	1.357	0.010
Xiantao	0.892	0.833	0.663	0.685	0.734	0.730	0.728	0.726	0.610	0.740	0.006
Qianjiang	0.788	0.765	0.659	0.573	0.585	0.569	0.593	0.622	0.672	0.645	0.006
Tianmen	1.070	1.073	1.046	1.022	1.136	1.217	1.218	1.200	1.168	1.129	0.006

Note: Due to space limitations, this paper chose to show the odd years and the average value of the cultivated land resource curse coefficient.

**Table 3 ijerph-18-00858-t003:** Results of the panel data model of the cultivated land resource curse in Jianghan Plain.

	No Cultivated Land Resource Curse Zone					Potential Cultivated Land Resource Curse Zone			Slightly Cultivated Land Resource Curse Zone								Severe Cultivated Land Resource Curse Zone	
	Caidian	Dangyang	Zhijiang	Yunmeng	Qianjiang	Yingcheng	Hanchuan	Xiantao	Jingshan	Shayang	Zhongxiang	Gongan	Shishou	Honghu	Songzi	Tianmen	Jiangling	Jianli
*FI*	0.041	0.137	0.188	−0.033	0.496	0.291	−0.045	−0.099	0.083	**1.113** **	**0.111** *	**0.245** *	0.715	0.592	**0.102** *	**0.208** **	**2.228** ***	**4.993** ***
*pro*	(0.697)	(0.526)	(0.415)	(0.759)	(0.207)	(0.202)	(0.881)	(0.765)	(0.356)	(0.040)	(0.099)	(0.100)	(0.372)	(0.256)	(0.195)	(0.016)	(0.001)	(0.009)
*UR*	0.062	−0.041	−0.062	0.007	−0.046	0.103	0.052	−0.028	0.035	0.013	0.030	0.307*	0.022	0.098	0.002	**0.042** *	0.012	**0.104** *
*pro*	(0.511)	(0.778)	(0.529)	(0.957)	(0.738)	(0.229)	(0.684)	(0.636)	(0.635)	(0.885)	(0.729)	(0.073)	(0.891)	(0.206)	(0.988)	(0.078)	(0.696)	(0.129)
*PCA*	0.062	−0.033	0.019	−0.028	−0.137	**0.331** *	**0.511** *	**0.213** *	**0.056** *	**0.072** *	**0.145** **	**0.035** *	0.031	**0.301** **	**0.078** ***	0.018	**0.339** **	**0.178** *
*pro*	(0.847)	(0.800)	(0.945)	(0.949)	(0.213)	(0.080)	(0.117)	(0.170)	(0.146)	(0.184)	(0.035)	(0.086)	(0.927)	(0.014)	(0.001)	(0.356)	(0.038)	(0.158)
*STGD*	−0.492	−0.240	−0.349	0.069	−0.107	**−0.296** **	**−0.004** *	**−0.396** **	0.109	0.006	0.171	**0.243** *	0.391	0.081	**0.356** **	**0.043** *	**0.958** ***	**0.734** **
*pro*	(0.511)	(0.285)	(0.403)	(0.814)	(0.770)	(0.019)	(0.189)	(0.036)	(0.513)	(0.974)	(0.487)	(0.130)	(0.490)	(0.605)	(0.015)	(0.035)	(0.000)	(0.014)
*GA*	−0.150	−0.176	−0.283	0.052	−0.110	**−0.325** *	−0.031	**−0.357** *	−0.148	0.200	−0.184	−0.536	−0.688	−0.256	−0.938	−0.006	**−3.814** ***	**−3.241** ***
*pro*	(0.278)	(0.319)	(0.336)	(0.823)	(0.739)	(0.054)	(0.927)	(0.153)	(0.561)	(0.659)	(0.608)	(0.382)	(0.518)	(0.418)	(0.416)	(0.990)	(0.000)	(0.006)
*GO*	0.027	−0.049	0.246	−0.049	−0.046	0.005	**−0.002** *	−0.154	−0.067	0.025	−0.064	0.020	−0.231	0.261	−0.302	−0.286	**−0.469** ***	**−0.199** *
*pro*	(0.895)	(0.678)	(0.567)	(0.592)	(0.661)	(0.952)	(0.178)	(0.375)	(0.518)	(0.837)	(0.711)	(0.903)	(0.266)	(0.321)	(0.373)	(0.405)	(0.000)	(0.151)
*TPM*	0.375	0.904	−4.896	−1.048	−7.639	−1.182	2.238	**−8.247** *	**−3.445** *	**−11.073** **	0.781	**−4.715** *	**−4.867** *	**−0.045** **	**−12.429** ***	**−8.401** *	**−15.737** ***	−7.108
*pro*	(0.942)	(0.869)	(0.362)	(0.927)	(0.286)	(0.842)	(0.665)	(0.099)	(0.158)	(0.021)	(0.857)	(0.120)	(0.098)	(0.040)	(0.002)	(0.164)	(0.000)	(0.449)
*Y* _1_	**−0.529** ***	**−0.115** *	**−0.200** **	**−0.154** *	−0.240	**−0.538** *	**−0.787** *	−0.276	**0.196** *	**1.014** *	**0.094** *	0.073	0.055	**0.112** *	0.188	**1.311** *	**1.505** **	**0.571** ***
*pro*	(0.001)	(0.055)	(0.018)	(0.125)	(0.382)	(0.140)	(0.115)	(0.275)	(0.134)	(0.129)	(0.074)	(0.321)	(0.265)	(0.141)	(0.374)	(0.131)	(0.039)	(0.001)

Note: *Pro* represents the probability corresponding to the variable coefficient; *** means *Pro* < 0.01 is very significant, ** means *Pro* < 0.05 is significant, * means *Pro* < 0.2 is generally significant. Results of significance were bolded for easy observation.

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
