# Peer review of "Spatio-Temporal Differentiation and Driving Mechanism of the “Resource Curse” of the Cultivated Land in Main Agricultural Production Regions: A Case Study of Jianghan Plain, Central China"

_ijerph, 2021, doi:10.3390/ijerph18030858_

Round 1
Reviewer 1 Report
The article deals with the topic of the 'resource curse' with a specific focus on the Chinese case, notably the Jianghan Plain. The topic is of utmost interest from the point of view of economic development and the conservation and reproduction of resources.
However the structure of the article presents some critical issues.
The abstract appears more like a concluding paragraph. There are no research questions, a general overview of the issues dealt with and the objectives underlying the research. It is advisable to rewrite the abstract leaving the results in the conclusions paragraph. Contents of lines 172 -177 may be useful in the abstract.
Introduction provides the general background. However, a deeper explanation of the content of the sessions is needed. Lines 102-105 cannot be a simple list of session titles.
The literature review is coherent and addresses both International and Chinese references. A table or a summary diagram related to the various approaches could enrich the article.
The materials and methods are well organized in paragraphs with maps, formulas and tables that help the reading of the following sessions.
Concerning some paragraphs of the 4 sessions, they appear quite consistent with the method described. In paragraph 4.2.2, please pay attention to the bullet points list. Recurring layout problems do not allow the understanding of some passages in the text.
The colors in Fig 4 are too similar.
The discussion, in drawing some policy directions, seems still too linked to the bibliography (this is not a literature review paragraph). Instead, it would be very useful to reinforce policy issues starting from the results of the case study.
Finally, concerning the structure, the 'discussion' session comes before the 'conclusions'. It is recommended to review the order and contents of the paragraphs.
There are some minor recurring writing problems.
Firstly, some points need to be properly refined according to scientific and technical language.
Secondly:
- in the numbered lists use the semicolon and not the full stop (for example lines 20, 22, 26, 91, 95, 97 (the same thing happens in the paragraph "conclusions")
- there are also some numbered paragraphs (279-317 and 455 - 487) in which it is not clear what the numbers at the beginning of the line refer to and how they are linked together.
In addition, the layout of sessions 5 and 6 require a review in terms of margins and line numbering.
Reviewer 2 Report
General comments. The paper is descriptive and not enough information. In the present paper the Authors analyze the effect of spatio-temporal differentiation and driving mechanism of the "Resource Curse" of the cultivated land in main agricultural production regions of Jianghan Plain. The research that you have performed is really interesting in my opinion, but manuscript is weak. Unfortunately, the work is not an experimental work. My main concerns with the paper are lack of detail of the experiment and also lacks a hypothesis
The present paper does not meet the scientific standard of the Journal, for a number of main concerns:
- Overall, the manuscript is so long. Needs to be rewritten and rearranged to shorten the whole manuscript.
- The abstract does not reflect the essence of the work.
- The keywords should be different than in the title.
- The manuscript is like a report and not an original research paper. Especially the Introduction – e.g. the text from line 54 to 82 - how is it related to work?
- Not enough evidences and literature data are presented to describe the importance of between the abundance of cultivated land resources and economic growth. Therefore, introduction is not enough informative. The authors have writeen ‘existing studies have mostly focused on the macro level’ (line 84 to 89) but where are literature positions ?
- Information is duplicated throughout the text line 89 - 97 and 172-176 and 221-226 and 264-274.
- The manuscript is confused and the reader does not have a clear view of the experimental plan: in mat & met section, the description of the experimental set-up is not clear, and the same is evident from the results.
- In the chapter material and methods it should be explain :
- What studies were used in the development of the formulas: 1, 2 3.
- Line195 - 199 there is no need to put city names simultaneously in the text and figure 2
- Line 203-210 please introduce credible data of economic development level e.t.c, this part of text is not enough informative e.g. ‘recent decades, the economic development of Jianghan Plain has lagged noticeably’
- Line 212- 214 should be entries in the reference list
- Line 324-327 should be in literature section
- line 391 - 393 should be in literature section
- line 445-453 should be in mat & met section
- Line 489 - 498 should be in mat & met section
- In the material and methods section, you do need to describe statistical method, not in the results Should be clearly explain a statistical process.
- The table 3. requires re-eding. The adopted concept of presenting the results in the table is not very clear.
- Discussion, page 16: point 1, 2, 3, 4 should be discussed from literature positions.
- Conclusions section standard should be in after discussion
- The text of conclusion section: ‘This study took 18 county-level …. panel regression method’ - was presented before and is not a summary.
- In the conclusion section, is very weak: for example “ From 2001 to 2017, the curse coefficient of cultivated land resources in Jianghan Plain in China generally showed a downward trend, but the gap between regions did not change much, and the curse phenomenon of cultivated land resources in large regions did not improve significantly.” - what research problem did the authors solve?
- The article is poorly written, eg. ‘scholars’ in all text. English is very weak and require a solid correction. The article is poorly prepared in editing requirements.
Regards
Round 2
Reviewer 2 Report
Comments to Author
I can now inform you that the Manuscript „
Spatio-temporal Differentiation and Driving Mechanism of the"Resource Curse" of the Cultivated Land in Main Agricultural Production Regions: A Case Study of Jianghan Plain, Central China” Authors analyze the effect of spatio-temporal differentiation and driving mechanism of the "Resource Curse" of the cultivated land in main agricultural production regions of Jianghan Plain.
The subject of the manuscript falls within the scope of the Section "Occupational Safety and Health" of IJERPH. The subject of the manuscript is interesting. The results could be considered as an original contribution. I have no objections to manuscript. Recommend publication in the journal.
Best regards